# Retrograde Analysis of Calcium Signaling by CaMPARI2 Shows Cytosolic Calcium in Chondrocytes Is Unaffected by Parabolic Flights

**DOI:** 10.3390/biomedicines10010138

**Published:** 2022-01-08

**Authors:** Andreas Hammer, Geraldine Cerretti, Dario A. Ricciardi, David Schiffmann, Simon Maranda, Raphael Kummer, Christoph Zumbühl, Karin F. Rattenbacher-Kiser, Silvan von Arx, Sebastian Ammann, Frederic Strobl, Rayene Berkane, Alexandra Stolz, Ernst H. K. Stelzer, Marcel Egli, Enrico Schleiff, Simon L. Wuest, Maik Böhmer

**Affiliations:** 1Institute for Molecular Biosciences, Johann Wolfgang Goethe Universität, 60438 Frankfurt am Main, Germany; hammer@bio.uni-frankfurt.de (A.H.); Ricciardi@bio.uni-frankfurt.de (D.A.R.); 2Space Biology Group, Institute of Medical Engineering, School of Engineering and Architecture, Lucerne University of Applied Sciences and Arts, 6052 Hergiswil, Switzerland; geraldine.cerretti@hslu.ch (G.C.); karin.rattenbacher@hslu.ch (K.F.R.-K.); marcel.egli@hslu.ch (M.E.); simon.wueest@hslu.ch (S.L.W.); 3National Center for Biomedical Research in Space, University of Zurich, Innovation Cluster Space and Aviation (UZH Space Hub), 8600 Dübendorf, Switzerland; 4Institute of Mechanical Engineering and Energy Technology, School of Engineering and Architecture, Lucerne University of Applied Sciences and Arts, 6048 Horw, Switzerland; david.schiffmann@hslu.ch (D.S.); simon.maranda@hslu.ch (S.M.); silvan.vonarx@hslu.ch (S.v.A.); sebastian.ammann@hslu.ch (S.A.); 5Institute of Electrical Engineering, School of Engineering and Architecture, Lucerne University of Applied Sciences and Arts, 6048 Horw, Switzerland; raphael.kummer@hslu.ch (R.K.); christoph.zumbuehl@hslu.ch (C.Z.); 6Buchmann Institute for Molecular Life Sciences (BMLS), Johann Wolfgang Goethe Universität, 60438 Frankfurt am Main, Germany; frederic.strobl@physikalischebiologie.de (F.S.); berkane@med.uni-frankfurt.de (R.B.); stolz@em.uni-frankfurt.de (A.S.); ernst.stelzer@physikalischebiologie.de (E.H.K.S.); 7Institute of Biochemistry 2, Johann Wolfgang Goethe Universität, 60438 Frankfurt am Main, Germany

**Keywords:** CaMPARI, articular chondrocytes, cytosolic free calcium, gravity, parabolic flight, high throughput screening

## Abstract

Calcium (Ca^2+^) elevation is an essential secondary messenger in many cellular processes, including disease progression and adaptation to external stimuli, e.g., gravitational load. Therefore, mapping and quantifying Ca^2+^ signaling with a high spatiotemporal resolution is a key challenge. However, particularly on microgravity platforms, experiment time is limited, allowing only a small number of replicates. Furthermore, experiment hardware is exposed to changes in gravity levels, causing experimental artifacts unless appropriately controlled. We introduce a new experimental setup based on the fluorescent Ca^2+^ reporter CaMPARI2, onboard LED arrays, and subsequent microscopic analysis on the ground. This setup allows for higher throughput and accuracy due to its retrograde nature. The excellent performance of CaMPARI2 was demonstrated with human chondrocytes during the 75th ESA parabolic flight campaign. CaMPARI2 revealed a strong Ca^2+^ response triggered by histamine but was not affected by the alternating gravitational load of a parabolic flight.

## 1. Introduction

Calcium (Ca^2+^) is a fundamental secondary messenger in all eukaryotic cells. Ca^2+^ signaling components integrate chemical and physical stimuli to drive developmental and physiological responses. Among these processes are Ca^2+^-dependent neurotransmitter release, muscle contraction, or the biological and biochemical processes downstream of membrane receptors, as in the IP3 pathways, which play a role in regulating cell division [1,2,3]. Ca^2+^ signals vary in their amplitude, temporal and spatial properties, and site of Ca^2+^ entry [1]. Since increased cytosolic Ca^2+^ concentrations can cause cell distress or even cell death [2], maintaining Ca^2+^ homeostasis is essential for living cells. Ca^2+^-transporters such as Ca^2+^-ATPases ensure a stable cytosolic Ca^2+^ concentration by pumping Ca^2+^ ions into the extracellular space or intracellular stores, mainly the endoplasmic reticulum (ER), from where they can be released into the cytosol upon signal activation [3]. 

Since dysregulation of Ca^2+^ homeostasis is related to many diseases, including skeletal muscle diseases, Alzheimer’s disease, and osteoporosis [4,5,6], a more profound understanding of the involvement of Ca^2+^ in these pathways could further medical drug development.

### 1.1. Genetically Encoded Ca^2+^ Indicators

A toolset of Ca^2+^-reactive dyes and genetically encoded Ca^2+^ indicators are available to study spatial and temporal properties of Ca^2+^ signaling in vivo [7]. One example among the fluorescent dyes is Fura-2, containing an 8-coordinate tetracarboxylic chelating site and stilbene chromophores. Upon Ca^2+^-binding, conformational changes increase the fluorescence intensity up to 30-fold [8]. In addition, genetically encoded Ca^2+^ indicators (GECIs) have been generated to measure intracellular Ca^2+^ concentrations in a non-invasive fashion. GECIs are engineered fluorescent proteins that change conformation in response to the cytosolic Ca^2+^ concentration [9]. 

GECIs such as the fluorescence protein-based Ca^2+^ indicators R-GECO and Cameleon provide a much stronger quantum yield and increased resolution for live-cell imaging [10,11]. R-GECO (red fluorescent genetically encoded Ca^2+^ indicator for optical imaging) is an intensiometric Ca^2+^ sensor whose fluorescence intensity increases in response to rising cytosolic Ca^2+^ concentrations. These GCaMP-class Ca^2+^ imaging proteins consist of a green fluorescent protein (GFP)- or red fluorescent protein (RFP)-derived fluorophore fused to calmodulin (CaM) and an M13 peptide. When Ca^2+^ binds to calmodulin, the two GFP or RFP halves come together to form a functional fluorescent protein [10,11].

The Cameleon-type Ca^2+^ reporters are ratiometric FRET (Förster resonance energy transfer) based indicators. An enhanced cyan fluorescent protein (ECFP) serves as the FRET donor, whereas a yellow fluorescence protein (YFP) works as a FRET acceptor [12]. With increasing Ca^2+^ concentrations, increased amounts of energy are transferred to the acceptor protein [12]. 

However, in both systems, the changes in fluorescence intensity or FRET are reversible, making it necessary to image samples in real-time to observe changes in cytosolic Ca^2+^ concentrations. Therefore, these spatial and temporal dependencies of observation and treatment limit the number of samples analyzed drastically.

### 1.2. CaMPARI

CaMPARI (calcium-modulated photoactivatable ratiometric integrator) is based on the fluorescent protein mEos that irreversibly converts from green to red emission when illuminated with near UV light at 405 nm [13]. In CaMPARI, a circular permutated Eos (cpEos) is attached to CaM and an M13 peptide. In contrast to mEos alone, photoconversion in CaMPARI is calcium-dependent, and efficient and irreversible green-to-red conversion only occurs when elevated intracellular Ca^2+^ and 405 nm illumination coincide [13]. The latest iteration, CaMPARI2, was designed with a two-fold increase in red fluorescence intensity, a higher Ca^2+^-binding rate, and a significantly increased red-to-green fluorescence ratio than the original CaMPARI [14]. In addition, several variants were developed with distinct affinities to cytosolic Ca^2+^. 

With CaMPARI, a decoupling of treatment and analysis is possible. The green-to-red conversion is caused by a covalent, and therefore stable fluorophore modification. The red-to-green ratio thus offers a snapshot of intracellular Ca^2+^ concentrations at the time of photoconversion and is only affected by the new synthesis of CaMPARI protein in its initial green form and the degradation of red CaMPARI. Considering these factors, the signal can remain constant for from hours to several weeks [15]. Furthermore, the simplicity of CaMPARI photoconversion at a distinct time point during an experiment enables the simultaneous investigation of Ca^2+^ concentrations in many samples. 

This decoupling is advantageous, especially for rather unconventional experimental conditions, where it is challenging to treat and measure cell culture simultaneously. One example of these experimental conditions are parabolic flights: The mechanical load on microgravity platforms makes conventional electrophysiological measurements, requiring precise micromanipulation, very challenging [16]. Therefore, in recent years, non-invasive optical instruments such as plate readers [17] or the FLUMIAS microscopes [18,19,20] have been used. For example, cells were stained with fluorescent dyes, such as Fura-2 or Fura-3, to measure changes in Ca^2+^ concentrations [17,21]. However, when using a plate reader or a microscope during a parabolic flight, the experiment hardware is exposed to accelerations, resulting in experimental artifacts if not adequately controlled [22]. 

In this work, we present a much more robust approach: we use multiple 405 nm LED arrays to convert CaMPARI2 from the green to the red fluorescent state in a large number of 96-well plates. As a marker of cytosolic Ca^2+^ concentration, the green-to-red conversion was measured after the flight using a ground-based Sartorius’ IncuCyte S3 high-throughput fluorescence microscope that allowed large-scale imaging of up to six 96-well plates in parallel. We used the human chondrocyte cell culture C28/I2 as a model for the experiments presented here. 

### 1.3. Chondrocytes and Histamine

Chondrocytes are the only living cells found in cartilage. They maintain healthy articular joints, e.g., by synthesizing and secreting the extracellular matrix and lubricants like hyaluronan and lubricin [23]. However, they are also involved in developing rheumatoid- and osteoarthritis [24,25,26,27,28,29]. Both forms of arthritis are characterized by degeneration and loss of articular cartilage. However, in osteoarthritis, the cause is mechanical wear on the joints, while rheumatoid arthritis is an autoimmune disease in which the body’s immune system attacks the body’s joints. 

Immunohistochemical staining identified an enhanced abundance of histamine H1 and H2 receptors and the histamine-producing enzyme histidine decarboxylase in a variable proportion of human articular chondrocytes in osteoarthritis cartilage specimens. Furthermore, histamine was found with concentrations of up to 210 nM in synovial fluid in patients with rheumatoid arthritis [30]. Histamine has a primary role in allergic and inflammatory reactions, but it also stimulates chondrocyte proliferation [31,32]. Taken together, this strongly suggests that histamine and histamine receptor expression in osteoarthritis cartilage is potentially a pivotal contributor to the aberrant hyperactive phenotype of osteoarthritis chondrocytes, including their ability to express pro-inflammatory cytokines and cartilage-degrading enzymes [32,33,34].

The binding of histamine to histamine receptors triggers an influx of Ca^2+^ into chondrocytes. However, the precise regulation of Ca^2+^ homeostasis in these cells is an essential factor for standard matrix metabolic and secretory functions [35,36]. A disturbance of Ca^2+^ homeostasis can contribute to chronic articular joint disease, such as arthritis and related inflammation [24,27,28,29,37]. The role of Ca^2+^ signaling in chondrocytes has recently been reviewed in detail [38]. In this work, we used treatment with histamine as a positive control stimulus to trigger intracellular Ca^2+^ elevation. 

In addition to biochemical stimuli, physical stimulation of chondrocytes leads to Ca^2+^ elevations in the cytosol [39]. For example, exposure of isolated chondrocytes to osmotic challenges [39,40,41,42,43,44,45,46], fluid flow [47], compression [48], deformation [49,50], and aspiration [51] have reportedly triggered increased cytosolic free calcium. Furthermore, mechanical stimulation and subsequent Ca^2+^ elevations increase the synthesis of extracellular matrix (ECM) proteins [52]. Therefore, chondrocytes have been used as model systems to study changes in intracellular Ca^2+^ concentrations to understand mechanotransduction pathways [16].

With an absence of mechanical load on the articular matrix, for example, in microgravity during spaceflights, the maintenance of the extracellular matrix is reduced. This effect is under investigation for manned space travel since it leads to osteoarthritis, dramatic degeneration of cartilage, and bone mass degradation [53]. A healthy cartilaginous matrix is necessary to improve astronauts’ health in the future, especially regarding potential long-term space travel missions. Therefore, further understanding of the molecular involvement of Ca^2+^ in these processes is required.

In intact cartilage, the onset of a Ca^2+^ elevation in response to physical stimuli is observable within 6–12 s [54], i.e., within the time frame of a single parabola during a parabolic flight. In this study, we implemented CaMPARI2, a genetically encoded Ca^2+^ indicator, and designed photoconversion hardware to measure changes in intracellular Ca^2+^ concentrations in chondrocytes during the different gravity phases of a parabolic flight and in response to histamine.

## 2. Materials and Methods

### 2.1. Vector Construction

The coding sequences of the five CaMPARI2 variants wt (Addgene #101060), F391W (Addgene #101061), L398T (Addgene #101064), F391WG395D (Addgene #101063), H396K (Addgene #101062) were amplified from the original pAAV vectors via PCR using primer pair CaMPARI2_attB1_KZ 5′-GGGGACAAGTTTGTACAAAAAAGCAGGCTGCCACCATGCTGCAGAACGAGCTTGC-3′ and CaMPARI2_attB2 5′-GGGGACCACTTTGTAC-AAGAAAGCTGGGTTTTATGAGCTCAGCCGACCTA-3′. PCR products were cloned into pDONR221 (Invitrogen) via BP reaction. The CaMPARI2 reporters were then transferred into the destination vector pMAX-DEST (Addgene #37631), using the Gateway^®^ LR Clonase^®^ II enzyme mix. The resulting pMAX-plasmids contained a CMV-promotor for eukaryotic cells, a Kozak sequence, a nuclear export signal, a His_6_-epitope tag, and a terminating SV40 polyA-tag.

### 2.2. Cell Culture and Treatment

The C28/I2 human chondrocyte cell line [55,56] as purchased from Merck (Schaffhausen, Switzerland) and maintained in T75 flasks containing Dulbecco’s Modified Eagle’s Medium (DMEM, w: 1.0 g/L Glucose, w: L-Glutamine, w: 25 mM HEPES, w: Sodium pyruvate, w: 3.7 g/L NaHCO_3_, PAN Biotech, Aidenbach, Germany) supplemented with 10% fetal bovine serum (FBS, Cytiva, Marlborough, MA, USA) and 1% PenStrep in a 37 °C incubator in a 5% CO_2_ atmosphere. Cells were subcultured in a 1:5 ratio (1 × 10^6^ cells/T75 flask) when they reached 80–90% confluence. The cells were harvested with Trypsin and washed with PBS. 

For treatments, cells were plated in 96-well plates (7000 cells per well) and cultured at 37 °C for 72 h. Then cells were exposed to a parabolic flight for the indicated times. Histamine-treated cells were exposed to 100 µM histamine for 15 s immediately after the parabolic flight.

### 2.3. Chondrocyte Transfection

C28/I2 chondrocytes were transfected using Lipofectamine 3000. A total of 7000 cells were seeded per well into eight 96-well-plates (µCLEAR^®^ BLACK 655090, Greiner Bio-One, Frickenhausen, Germany) for every flight day, 24 plates in total. After 24 h in a 5% CO_2_ atmosphere at 37 °C, each well was treated according to the manufacturer’s protocol with 200 ng of DNA, 0.4 µL of P3000 reagent, and 0.6 µL of Lipofectamine 3000 reagent. The transformed vectors were the three constructs pmaxGFP (Lonza), pMAX-DEST CaMPARI2 [K_D_ = 199.2] and pMAX-DEST CaMPARI2-F391W [K_D_ = 109.7]. Cells in each well were incubated with 200 µL DMEM medium and the transfection reagents for 44 h before the flight day.

### 2.4. Parabolic Flight Experimental Design

Parabolic flights are among the most used ground-based platforms in gravitational research that expose samples to real microgravity [57]. The experimental design for the parabolic flight experiments consisted of three replicate flight days, all flown during the 75th ESA parabolic flight campaign in Bordeaux, France. Each flight day comprised 31 consecutive parabolas. A single parabola comprises 5 flight phases (Figure 1); the Pre-Parabola and Post-Parabola phases just before and after the parabola at 1 g, the Pull-Up and Pull-Out phases at the beginning and end of the parabola at around 1.8 g each, and the Zero-G phase at around 10^−5^ g. Each parabola offers about 20 s of hypergravity (Pull-Up phase), followed by ca. 22 s of microgravity (Zero-G phase), and concluded by another 20 s of hypergravity (Pull-Out phase) (Figure 1). The 96-well plates with chondrocytes were kept inside the flight hardware at 37 °C in an ambient atmosphere and were illuminated by LED light for photoconversion during parabolas 0, 10, 20, and 30.

### 2.5. Parabolic Flight Hardware

Novel custom-made hardware consisting of four identical functional units was developed to guarantee optimal cell culture condition and precise synchronization of the photoconversion with the parabolic maneuvers (Figure 2). Two 96-well plates were placed in each unit. Each well was irradiated using an LED array placed below the 96-well plate at the indicated time point. A black spacer between the individual LEDs of the LED array and the usage of black 96-well plates ensured homogeneous illumination and prevented cross irradiation of neighboring wells. The LEDs (ATDS3534UV405B, Kingbright, New Taipei City, Taiwan) had a peak emission at 405 ± 5 nm and ca. 1.1 W radiant flux. However, the LEDs’ power dissipation of ca. 2.8 W (per LED) required thermal management to prevent overheating of the cells (described below). 

The functional unit structure (Figure 2) was mainly manufactured from 3D-printed acrylonitrile butadiene styrene (ABS) and was tailored to the 96-well plate to avoid undesired plate movement during the flight. The sterile 96-well plates were sealed before integration, placed in the hardware, and secured with a lid and elastic spring elements. The LEDs were mounted on a single custom-made printed circuit board (PCB) with an aluminum core, integrated into the 3D-printed housing. A second custom-made PCB featured a microcontroller (tinyK22 designed by the Lucerne School of Engineering and Architecture, Horw, Switzerland), and interfaced with peripheral devices such as LEDs and sensors. The microcontroller controlled all components in the unit, provided a serial communication interface to an external board computer (laptop), and logged various housekeeping data, including acceleration, temperature, air pressure, supply voltage, and power consumption. Just below the PCB with the LEDs, a thermal buffer was integrated. Finally, the units were placed in air-tight aluminum containers for safety reasons to avoid environmental contamination in case of media leakage.

After the units were prepared and closed in the laboratory, they were mounted into a flight rack screwed to the aircraft floor. The air-tight aluminum containers and the flight rack were reused from a previous parabolic flight campaign [58]. In brief, the rack was built according to the requirements of Novespace (Bordeaux, France), and could provide space for up to 12 units. The units were packed in polyurethane foam to reduce mechanical vibration and ensure thermal insulation. Custom-made software was executed from a laptop, which controlled all units and measured acceleration during the flight to trigger the photoconversion synchronously with the parabolic maneuver. 

### 2.6. Thermal Management

A heating element was integrated into the spacer between the 96-well plates and the LEDs to stabilize the samples’ temperature at 37 °C (Figure 2). This was realized with a custom-made PCB manufactured on thermally well-conducting aluminum featuring a resistor array and a temperature sensor. Heating power was regulated by the microcontroller using the temperature sensors as feedback. The experiment had to be powered off during take-off and landing for safety reasons. Therefore, the units were packed in foam to guarantee sufficient thermal insulation during this period.

Each LED dissipated around 2.8 W when illuminated. In our experiment, per treatment condition and parabola, 11 wells were illuminated in rapid succession for 8 s each. Per unit, four LEDs were illuminated in parallel at the same time. Therefore, per unit and parabola, the 44 LEDs generated ca. 986 J of heat energy within less than 2 min. However, the insulation around the units prevented the fast dissipation of this excessive heat to the aircraft’s cabin. Hence, a thermal buffer was placed below the LED array to prevent overheating the samples. The buffer consisted of paraffin wax (Eicosane; Merck, Schaffhausen, Switzerland), with a melting point at ca. 36 °C, enclosed in an aluminum casing. It stabilized the temperature two-fold. First, during take-off and landing, it helped slow down cooling in combination with the insulation. Second, the buffer absorbed the heat generated by the LEDs during illumination. For this reason, the LEDs were mounted on an aluminum PCB, and the buffer’s casing was also made of aluminum and was mounted in close contact with the LED’s PCB. This design ensured that the heat energy from the LEDs would be quickly conducted away and absorbed by the paraffin wax.

### 2.7. Before the Flight

Four hours before the flight, chondrocytes were washed with 200 µL PBS. Then DMEM containing one of the drugs listed in Table 1 was added to each 96-well plate until a meniscus of liquid formed above each well (380–385 µL) to avoid all physical interference of air bubbles during the flight. To ensure the absence of air bubbles and prevent leakage during the flight, the plates were sealed with adhesive film (Greiner Bio-One™ EASYseal™, 676001) and placed inside the preheated hardware.

### 2.8. During the Flight

During parabolas 0, 10, 20, and 30 of each of the three flights, individual wells were illuminated for 8 s with photoconversion light. First, one well transfected with turbo-GFP was illuminated, followed consecutively by ten wells of CaMPARI2 or CaMPARI2-F391W, respectively. During normal gravity, two wells were illuminated, followed by two wells during the Pull-Up phase, two wells during the Zero-G phase, two wells during the Pull-Out phase, and finished by two wells at normal gravity Post-Parabola (Figure 1).

### 2.9. After the Flight

The seal’s adhesion was checked after the flight, and no air bubbles or leakages were detectable. For each parabola-set of cells, one well served as a positive control for photo conversion of CaMPARI2 and CaMPARI2-F391W with 100 µM histamine after the flight. The medium of these eight wells per plate was discarded directly after the flight, and DMEM cell culture medium containing 100 µM histamine was added to the wells. After 15 s, these wells were illuminated with the flight hardware for 8 s, and the wells were filled up with medium to 380 µL again. Subsequently, phase contrast and fluorescence images in the green and red channels were acquired. All plates were scanned on-site with an IncuCyte S3 microscope equipped with an IncuCyte Zoom 10 Plan Fluor objective (Sartorius, Göttingen, Germany). For each well, five non-overlapping regions were imaged.

### 2.10. Image Analysis and Statistics

For the analysis of the images taken by the IncuCyte S3 microscope, the basic analyzer of the IncuCyte 2021A software (Sartorius, Göttingen, Germany) was used. The optimal settings were determined based on 50 individual representative images as a sample for the 11,520 total images the IncuCyte S3 microscope took during the campaign. To detect green and red fluorescent cells (CaMPARI green and CaMPARI red), the segmentation adjustment was set to 1.0, and Top-Hat segmentation edge sensitivity was set to −30. A cleanup filter for hole fill of the non-fluorescent nuclei with 200 µm^2^ was applied, and an area filter of 150–5500 µm^2^ led to the recognition of objects only in the specified size range. The cell radius for both green and red fluorescent objects was set to 25 µm. The threshold for green computational units (GCU) was set to 4, and for the weaker red fluorescent signal of CaMPARI2, the red computational unit threshold (RCU) was set to 0.3. Finally, four different analysis metrics were calculated to quantify photoconversion in the C28/I2 cells (Table 2). 

These four analysis metrics were graphed and statistically evaluated using in-house R scripts. In addition, the quantitative image analysis metrics were matched with acceleration data, plate layouts, inhibitors, voltage supply, parabola numbers, temperatures, and measured acceleration. A repository with the R scripts (release 1.0) and the quantitative results from the IncuCyte 2021A software is available from GitHub (https://github.com/ZeroG-lab/PFCaMPARI; accessed on 5 January 2022). Finally, statistical analyses were performed using ANOVA and Tukey post-test. 

## 3. Results

### 3.1. Selection of CaMPARI2-Constructs

Expression and photoconversion of five CaMPARI2 constructs (CaMPARI2-F391W, CaMPARI2, CaMPARI2-H396K, CaMPARI2-F391W-G395D, and CaMPARI2-L398T) were tested after transient transformation into C28/I2 human chondrocytes to select CaMPARI2 variants with adequate photoconversion rates in response to a positive stimulus. As a result, all five variants could be expressed, and photoconversion was detectable after a 15-s treatment with 100 µM histamine and 4 s of 405 nm photoconversion light (Figure 3). Notably, the fluorescence intensity of the CaMPARI2-H396K variant was overall reduced, and the photoconversion rate of this variant was lower than expected based on the previously determined in vitro K_D_-values [14].

CaMPARI2-F391W and CaMPARI2, on the other hand, showed high photoconversion rates with 65% and 55%, respectively, without reaching saturation. Moreover, in non-elicited cells, these variants displayed 13% and 2% photoconversion rates, respectively (Pre-Parabola condition, Figure 4), indicating an already high sensitivity to basal calcium. These results are consistent with published basal cytosolic Ca^2+^ concentration in mammalian chondrocytes between 100 and 200 nM [67]. Furthermore, with K_D_ values of 199.2 nM for CaMPARI2 and 109.7 nM for CaMPARI2-F391W, the Ca^2+^ affinities of both variants are also higher than those of Fura-2-AM at 285 nM [68]. 

Next, C28/I2 cells were transfected with the CaMPARI2-F391W-G395D construct to determine the optimal illumination duration. This construct was used due to its low Ca^2+^-affinity, which allowed for longer illumination times and made an analysis of a photoconversion time course up to 32 s possible (Figure 3). 

CaMPARI2-F391W-G395D expressed in the C28/I2 human chondrocyte cell underwent visible photoconversion already after 1 s of illumination. After that short time, about 50% of the maximum possible red fluorescence after illumination was reached. However, the transfection of cells with Lipofectamine 3000 also led to dead cell artifacts when measuring red fluorescence due to the transfection agent’s cytotoxicity (Figure 1A, cells without histamine treatment after 16-sec illumination). Therefore, the photoconversion time was set to 8 s for the parabolic flight experiments to gain an optimal signal-to-noise ratio. Furthermore, using the variants CaMPARI2 and CaMPARI2-F391W, which have the highest affinity to Ca^2+^, for the parabolic flight will further increase the number of converted cells and improve the signal-to-noise ratio.

### 3.2. Gravi-Elicited Ca^2+^ Elevation Changes during a Parabolic Flight

To investigate gravity-elicited changes in cytosolic Ca^2+^-concentrations in chondrocytes during the 75th ESA parabolic flight campaign, novel hardware was designed to perform the photoconversion of CaMPARI2 during the flight (Materials and Methods, Section 2.5). Before each flight, the functional units were preheated to 37 °C overnight. In the morning, 2–3 h before take-off, the 96-well plates containing the C28/I2 human chondrocytes transfected with CaMPARI2-F391W and CaMPARI2 were treated with inhibitors (s. Table 1) and transferred into the functional units. Those were then installed and connected to the flight rack (Figure 2).

One hour after take-off, for approximately 1.5 h, the cells were exposed to a total of 31 parabolas, each consisting of three phases, a 20 s hypergravity-phase (Pull-Up) at ca. 1.5–1.8 g, a 22 s microgravity-phase (Zero-G), and another 20 s hypergravity-phase (Pull-Out) (Figure 1). During parabolas 0, 10, 20, and 30, individual wells were illuminated for 8 s, each, with photoconversion light to record a time course of Ca^2+^ elevation changes during the different g-levels. 

Pre-Parabola, two wells of cells expressing GFP, followed by two wells each of cells expressing CaMPARI2 and CaMPARI2-F391W, were sequentially illuminated. Subsequently, two wells of each CaMPARI2 construct were illuminated during the Pull-Up phase, two wells during the Zero-G phase, two wells during the Pull-Out phase, and finished by two wells at normal gravity Post-Parabola (Figure 1). Finally, one hour after the last parabola, the plane returned to Bordeaux (France), and the cells were immediately transported to the laboratory.

The hardware performed as intended. The brief temperature overshot during illumination, measured at the heating elements in contact with the samples, was ≤0.5 °C in all units and flights. 

As a positive control for cell fitness and the general ability of CaMPARI2 and CaMPARI2-F391W for photoconversion, one well per parabola and construct (8 wells per plate in total) were treated with 100 µM histamine for 15 s and illuminated for 8 s by 405 nm photoconversion light (Figure 3).

The IncuCyte S3 microscope then imaged all eight 96-well plates. After applying the mask created in the IncuCyte 2021A software (Materials and Methods, Section 2.7), photoconversion of CaMPARI2 and CaMPARI2-F391W in each well was quantified (Figure 5). 

The fluorescence intensity-based metrics were significantly less reliable than the conversion rate due to microscopy artifacts, including dead cells and other debris with strong red autofluorescence (Appendix A). Nevertheless, despite slight variations in the resulting analysis between the four metrics, the inhibitors showed similar effects regarding the conversion of the different CaMPARI2 constructs.

### 3.3. Construct and Flight-Specific Conversion Rate

When comparing the two constructs used for the transfection of C28/I2 cells, the differences between CaMPARI2 and CaMPARI2-F391W are noticeable (Figure 4). During the parabolas, the conversion rate of CaMPARI2 transfected cells is very low. Only 1–3% of cells show the conversion to red. The histamine control shows a conversion rate of 40%. CaMPARI2-F391W, on the other hand, shows a basal conversion rate of 10 to 15% during the different flight maneuvers. The histamine-treated cells transfected with CaMPARI2-F391W show a 45 to 60% conversion rate. C28/I2 cells were also transfected with GFP as a negative control for photo conversion since GFP cannot convert to red fluorescence under the illumination of 405 nm light. Both CaMPARI2 constructs indicate no significant change in cytosolic Ca^2+^ concentration during different parabolic flight phases. When taking a closer look at the conversion rate of 96-well plates with the different treatments of cells transfected with CaMPARI2-F391W during each flight phase (Figure 6), there is also no significant increase or decrease of cytosolic Ca^2+^ compared to the Pre-Parabola phase visible. Similar results were visible for the CaMPARI2 construct, albeit the base conversion rate was much lower, corresponding to the results described in Figure 4 (Appendix A). However, slight deviations can be detected by comparing the overall base conversion of differently treated 96-well plates. Compared to the untreated 96-well plate, the plates treated with GSK2193874 and Thapsigargin 5 h before the parabolas contained 5–10% more converted cells during all phases of the parabolic flight.

### 3.4. Inhibitor Specific Conversion Rate

To compare the different conditions of the eight 96-well plates used for the parabolic flight campaign, the after-flight positive control of CaMPARI2-F391W with histamine was analyzed (Figure 7). The cells were treated with 100 µM histamine directly after each flight. After 15 s of incubation, the cells were illuminated for 8 s with 405 nm photoconversion light. The conversion rate of the Pre-Parabola phase of each inhibitor was subtracted from the conversion rates to account for differences in photoconversion by basal Ca^2+^ levels (compare Figure 6 and Figure 7). Untreated samples and samples with DMSO-control, gadolinium chloride, and ruthenium red had comparable conversion rates. GSK2193874 reduced the conversion rate of C28/I2 cells by around 50% compared to untreated cells. Thapsigargin, flunarizine, and BAPTA completely blocked photoconversion of CaMPARI2-F391W.

### 3.5. Z’-Factor Analysis

The Z’ factor is widely used to measure the suitability of an assay for high-throughput analyses [69], and was calculated here to determine the efficiency of CaMPARI2-F391W in high throughput Ca^2+^ concentration measurements, particularly in the context of inhibitor screening. C28/I2 cells from all four parabolas of each flight, treated with 100 µM histamine alone, were compared to those cells pretreated with 10 µM BAPTA-AM and subsequently with 100 µM histamine. Z’-factors of 0.72, 0.49, and 0.75 were calculated, respectively (Figure 8, Appendix A), which indicates an excellent separation of positive and negative control samples. The Z’-factors were calculated for each flight separately since, for every flight, a different batch of cells was used and was transfected on different days, making it a different assay.

## 4. Discussion

Ca^2+^ quantification techniques like electrophysiology and imaging-based techniques like fluorescence microscopy and plate reader-based assays suffer from hardware artifacts when operating under conditions with varying acceleration during a parabolic flight [22]. Sophisticated controls and mathematical models are needed to decipher the actual gravity-elicited Ca^2+^ changes and control for hardware artifacts [22], further minimizing the throughput of such techniques. With CaMPARI2, we introduce a new reporter for Ca^2+^ quantification to microgravity experiments. Due to its retrograde nature, the quantification takes place post-treatment. Therefore, the quantification does not suffer from hardware artifacts due to changes in g-levels, and hardware can be much cheaper and smaller than with previous techniques, allowing for a much higher throughput of experimental conditions. Furthermore, compared to a plate reader, cells transfected with CaMPARI2 combined with a high-throughput microscope, like the Incucyte S3, offer single-cell resolution with a much better signal-to-noise ratio than plate readers.

It has been suggested that changes in gravity affect membrane fluidity, thereby leading to altered gating properties of mechanosensitive channels [16]. Chondrocytes are prime examples of cells that perceive mechanical signals from their environment and modulate their metabolic activity accordingly [22]. It has also been shown that mechanical stimulation in these cells using a glass micropipette elicits an immediate and transient increase in intracellular Ca^2+^ levels [54]. Furthermore, chicken chondrocytes have been treated with clinorotation at 30 rpm for up to 48 h. The intracellular Ca^2+^ concentration was measured using Fura-2/AM to test for the effect of microgravity on chondrocytes [70]. Ca^2+^ levels decreased in control cells at normal gravity and in clinorotated cells, but the decrease was slightly more pronounced in the clinorotated cells. However, the variations between experiments were substantial, questioning the statistical significance of the results. 

An increase of intracellular Ca^2+^ in response to changes in gravity was also detected using another cell line and the Fura-3-AM dye. The relative fluorescence of the dye in SH-SY5Y cells increased within 5 s by 0.8% upon clinorotation at 60 rpm, and by 0.7% upon transition from normal gravity to microgravity in a parabolic flight, with a 1.6% decrease in the hypergravity phase [21]. Interestingly, in another study, the same cells showed a 2% increase in hypergravity and a 2% increase in the following microgravity and hypergravity phases [17]. These contrasting results must be further put into perspective, considering that histamine-induced intracellular calcium levels increased by 300% in our positive control. This magnitude of the increase is more in line with what could be expected from a biological stimulus, including what has been measured for mechanical stimulation in chondrocytes [54]. Therefore, this study aimed to determine if acceleration changes during a parabolic flight can elicit a cytosolic Ca^2+^ increase in single cells, particularly in chondrocytes, and if so, what signaling pathways and components might be involved in this response.

Using histamine as a cytosolic Ca^2+^ elicitor with a known biological function, we selected CaMPARI2 variants with suitable Ca^2+^ affinities. The cytosolic Ca^2+^ concentration in mammalian cells ranges between 100 and 200 nM [67]. Histamine increases the cytosolic Ca^2+^ concentration for several seconds up to 1 mM [71]. Therefore, CaMPARI2 and CaMPARI2-F391W appeared the most suitable variants for working with mammalian cell culture because of their ability to measure small changes in intracellular Ca^2+^ concentration. Using CaMPARI2-F391W as an example, we could show that the high affinity to Ca^2+^ already leads to a conversion of 10% of cells transfected with CaMPARI2-F391W without any stimulus when illuminated for 8 s with 405 nm light (compare Figure 4). Therefore, even small increases in cytosolic Ca^2+^ concentrations during this 8 s of illumination would lead to a visible increase in conversion. However, based on the presented data, we conclude that C28/I2 human chondrocyte cells do not respond with Ca^2+^ elevation changes to changes in gravity in the observed range of 0–1.8 g. These results align with a parallel experiment, also using C28/I2 cells, performed in the same parabolic flight using a plate reader, two different Ca^2+^ dyes, and fixed control cells [22]. It would, however, be beneficial to perform further experiments with different cell types to accommodate for the different repertoire of ion channels, transporters, and receptors in these cells. Also, platforms that offer longer timeframes of hyper- and microgravity and cleaner gravity phases without prior hypergravity treatment would benefit such studies. Once a cell line is found that displays changes in Ca^2+^, CaMPARI2 can be used to perform full-scale pharmacological screens and threshold experiments to identify the level of gravity required to elicit changes in Ca^2+^ and the underlying signaling pathways.

This manuscript already presents a small example of such a pharmacological study. Using different Ca^2+^ and histamine pathway inhibitors, we identified crucial transporters involved in the histamine signaling and subsequent cytosolic Ca^2+^ increase (Figure 9). Two of the inhibitors, Gd^3+^ and Ruthenium red, did not significantly affect the cytosolic Ca^2+^ increase after histamine stimulus. Since both chemicals are described as inhibitors for Transient receptor potential vanilloid 4 (TRPV4) [72,73], and in the case of Gd^3+^ also as an inhibitor for stretch-activated ion channels (SACs) [74], we can conclude that these channels are not involved in the histamine-induced Ca^2+^ increase. On the other hand, GSK2193874, a specific inhibitor for the TRPV4 channel (Figure 9) [65], reduced the photoconversion of CaMPARI2-F391W compared to the untreated C28/I2 cells by 50% (compare Figure 7). However, GSK2193874 was described to have an off-target effect on the voltage-gated Ca^2+^ channel CACNA1C with an IC_30_ value of 10.5 µM. Therefore, this result could indicate that voltage-dependent Ca^2+^ channels may be involved in histamine signaling. 

Thapsigargin, a Sarco/endoplasmic reticulum Ca^2+^-ATPase (SERCA) inhibitor [75], caused complete inhibition of CaMPARI2 photoconversion, indicating the ER’s primary role in histamine-induced Ca^2+^-elevations [76]. After inhibition, the SERCA cannot transfer Ca^2+^ ions into its main store, the endoplasmatic reticulum (ER), which depletes Ca^2+^ ions after a few hours, preventing histamine-induced cytosolic Ca^2+^ increases originating from ER stores. Flunarizine, an antihistamine and Ca^2+^ entry blocker [77], also fully inhibited a histamine-induced increase in cytosolic Ca^2+^ (Figure 7). Flunarizine is a blocker of the H1R histamine receptor, suggesting that histamine-induced Ca^2+^-elevations originate from H1R. It is also known to block voltage-gated calcium channels and might exert a dual inhibitory function in histamine signaling [62,63]. However, further genetic or pharmacological studies will be necessary to confirm this hypothesis. Using CaMPARI2 as a high throughput analysis tool, large-scale pharmacological screens can be used to identify components involved in histamine signaling upstream of cytosolic Ca^2+^ increase. 

In summary, we successfully implemented a high-throughput analysis of cytosolic Ca^2+^ concentration using CaMPARI2 photoconversion as a readout. This system is suitable for environments with varying accelerations and can be used for large-scale pathways analysis with pharmacological libraries in the future. 

## Figures and Tables

**Figure 1 biomedicines-10-00138-f001:**
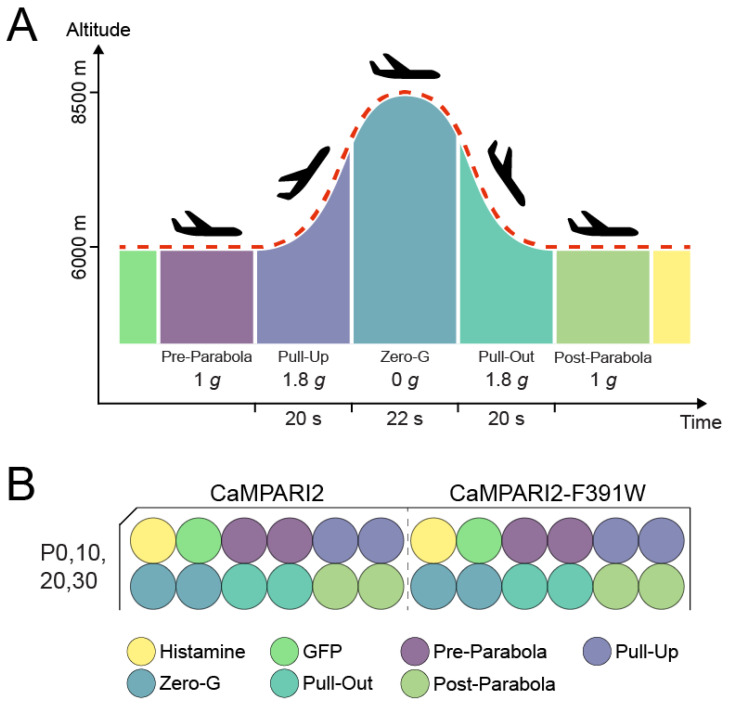
Graphical scheme of the parabolic flight experiment. (**A**) Scheme of the parabolic flight showing the different flight phases and acceleration. (**B**) Transfection-layout of the 96-well plates with C28/I2 cells. In the left half of the 96-well plate, the cells were transfected with CaMPARI2, while in the right half the cells were transfected with CaMPARI2-F391W. In each half, one LED could be activated at the same time. In each depicted parabola, the first wells illuminated were the GFP (green) transfected cells, followed by the first well of the Pre-Parabola phase, etc. Every well was illuminated for 8 s with 405 nm LEDs with 1.1 W radiant flux.

**Figure 2 biomedicines-10-00138-f002:**
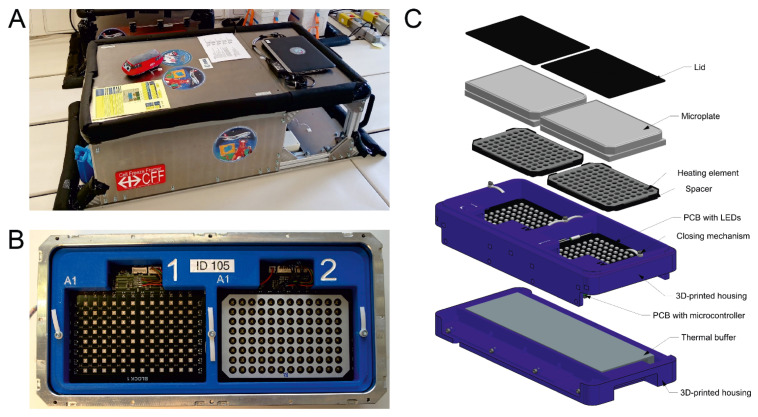
Parabolic flight hardware. (**A**) Flight rack mounted inside Novespace’s aircraft. (**B**) Top view on the functional units, showing the empty payload bays for the two 96-well plates. The spacer with the heating element was removed in the left bay, exposing the LED array. (**C**) Explosion view of the hardware, highlighting the most relevant elements.

**Figure 3 biomedicines-10-00138-f003:**
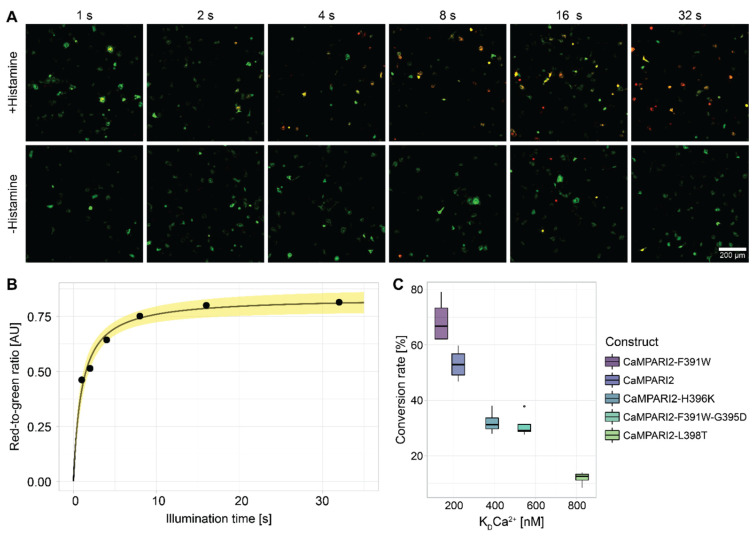
Time course of photoconversion illumination over 32 s and conversion rates of different CaMPARI2-variants in C28/I2 chondrocyte cells in response to histamine treatment. C28/I2 cells transfected with Lipofectamine 3000 and 100 ng of pMAX_CaMPARI2-F391W-G395D plasmid DNA were treated with 100 µM histamine and illuminated for different durations with 1.1 W 405 nm LEDs. (**A**) C28/I2 chondrocytes imaged by a Zeiss LSM780 laser scanning confocal fluorescence microscope. Scale bar = 200 μm. Green signal: excitation wavelength 488 nm, Argon 25 mW, detection range 490–588 nm, laser intensity: 3%; red fluorescence: excitation wavelength 561 nm, DPSS 561 10 mW, detection range 578–696 nm, laser intensity: 10%; Objective: Plan-Apochromat 10x/0.3 M27; Pinhole size: 2 AU; Pixel pitch: 1.66 µm; Pixel dwell time: 13 µs; Averaging: (8) Red and green fluorescence signals were merged to show the conversion ratio of green to red CaMPARI2. (**B**) The average intensity of red fluorescence divided by green fluorescence fitted with a saturation curve. The yellow area indicates the 90% confidence interval of the saturation curve. (**C**) Cells of the C28/I2 human chondrocyte line were transfected with five different constructs of CaMPARI2 using Lipofectamine 3000. Two days after the transfection, the cells were treated with 100 µM histamine and incubated for 15 s, followed by a 4 s long 405 nm photoconversion light from LEDs with 1.1 W radiant flux. The conversion rate is defined as the percentage of CaMPARI2-transfected cells showing the conversion from green to red fluorescence divided by the total number of fluorescent cells.

**Figure 4 biomedicines-10-00138-f004:**
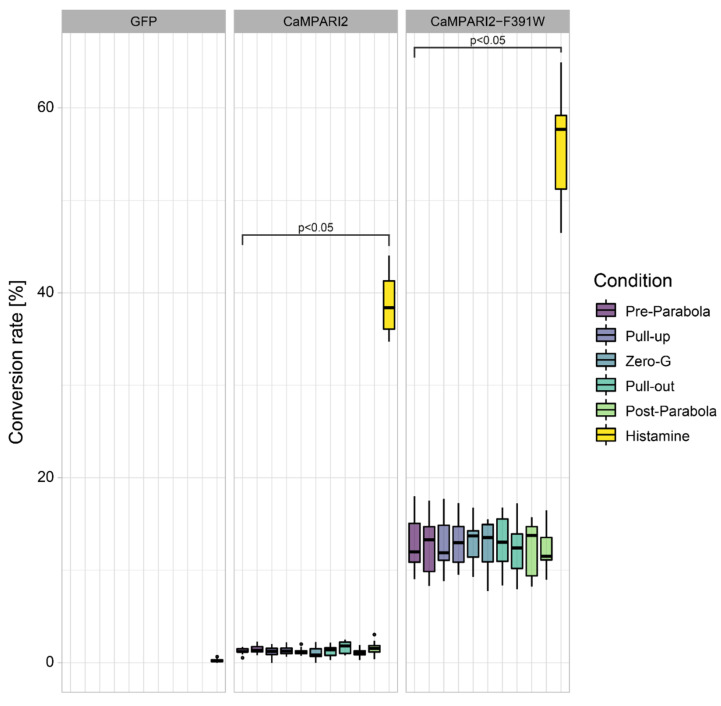
Conversion rates of C28/I2 human chondrocyte cells expressing GFP, CaMPARI2, or CaMPARI2-F391W. The cells were illuminated for 8 s with 405 nm light during the indicated flight phase for conversion. The cells expressing GFP were illuminated directly before each Pre-Parabola phase. For each parabola and CaMPARI2-construct, eight wells per 96-well plate were treated with 100 µM histamine for 15 s and subsequently illuminated for 8 s. The conversion rate was calculated using the mask described in 2.10. Brackets indicate significant differences compared to the Pre-Parabola phase (*p* < 0.05 using ANOVA and Tukey posthoc tests).

**Figure 5 biomedicines-10-00138-f005:**
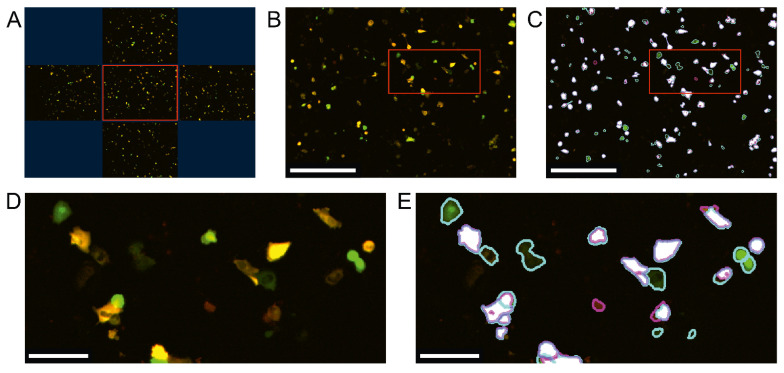
High throughput fluorescence microscopy workflow of C28/I2 human chondrocyte cell line transfected with CaMPARI2-F391W. (**A**) The IncuCyte S3 high-throughput microscope of Sartorius took five images per well of 96-well plates. The shown cells were treated with 100 µM histamine directly after the parabolic flight and were illuminated for 8 s by photoconversion light 15 s after the histamine stimulus. The centered image (red frame in (**A**)) was used as an example to show the mask created with the IncuCyte 2021A software in (**B**,**D**). (**B**,**D**) Green and Red fluorescence of CaMPARI2-F391W is depicted as an orange merge. The red frame is the image section displayed in (**C**,**E**). (**C**,**E**) Cells only identified as green fluorescent are circled cyan. Cells only identified as red fluorescent are circled purple. Cells identified by the IncuCyte mask as green and red fluorescent are colored white. The scales in (**B**,**C**) represent 500 µm; the scales in (**D**,**E**) represent 100 µm.

**Figure 6 biomedicines-10-00138-f006:**
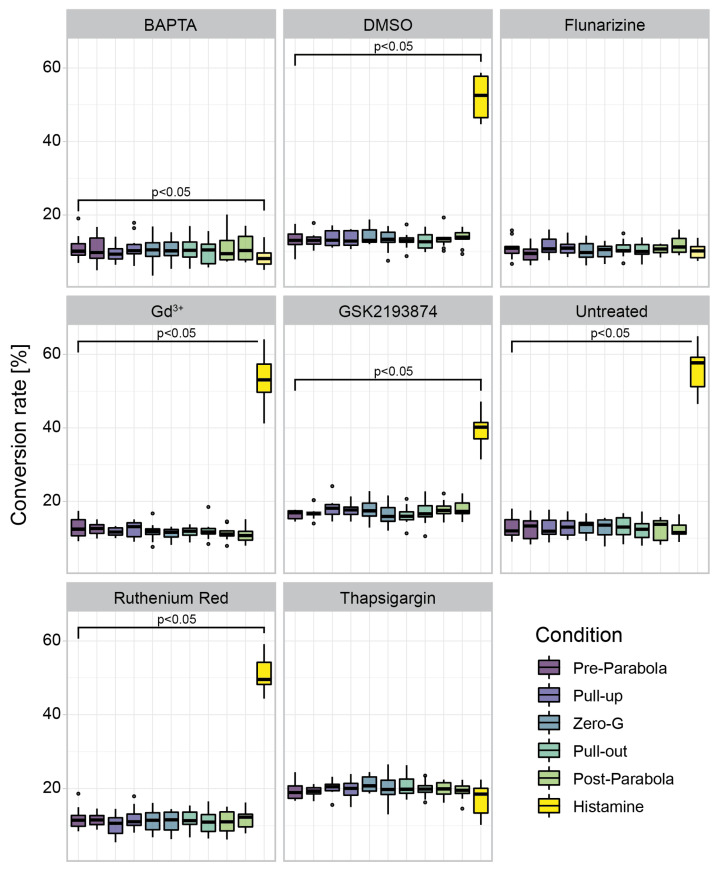
Conversion rates of cells transfected with CaMPARI2-F391W and subjected to a parabolic flight. Eight 96-well plates were used for each of the three parabolic flights. Each plate was treated with one of the displayed drugs 5 h before photoconversion. The cells were illuminated over four parabolas (0, 10, 20, and 30). A conversion rate was calculated for each flight phase and the histamine post-flight control (*n* = 12, 3 flights, 4 parabolas) using the IncuCyte 2021A mask described in 2.10. Brackets indicate significant differences compared to the Pre-Parabola phase (*p* < 0.05 using ANOVA and Tukey posthoc tests).

**Figure 7 biomedicines-10-00138-f007:**
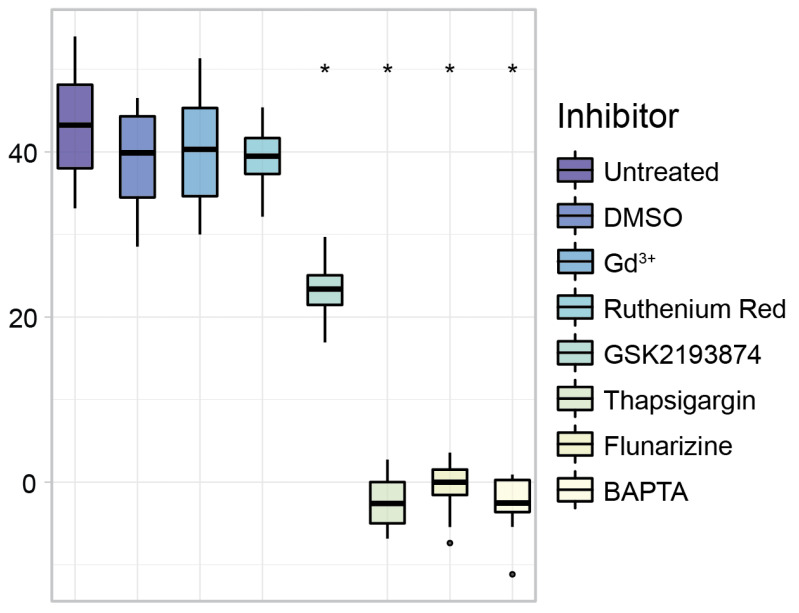
Conversion rates of C28/I2 cells transfected with CaMPARI2-F391W and treated with selected drugs. Immediately after the parabolic flights, cells were treated for 15 s with 100 µM histamine and subsequently illuminated for 8 s with 405 nm light. Asterisks indicate significant change compared to Untreated (*p* < 0.05 using ANOVA and Tukey posthoc tests). Untreated, DMSO-control, gadolinium chloride, and ruthenium red had no significant effect on the conversion rate of CaMPARI2-F391W after treatment with histamine. GSK2193874 reduced the conversion efficiency by 50% compared to untreated cells, whereas Thapsigargin and Flunarizine completely inhibited the conversion of CaMARI2-F391W. The conversion rate of the Pre-Parabola phase as a base conversion of each inhibitor was subtracted from the conversion rates. Asterisks indicate significant differences compared to cells only treated with Histamine (untreated) (*p* < 0.05 using ANOVA and Tukey posthoc tests).

**Figure 8 biomedicines-10-00138-f008:**
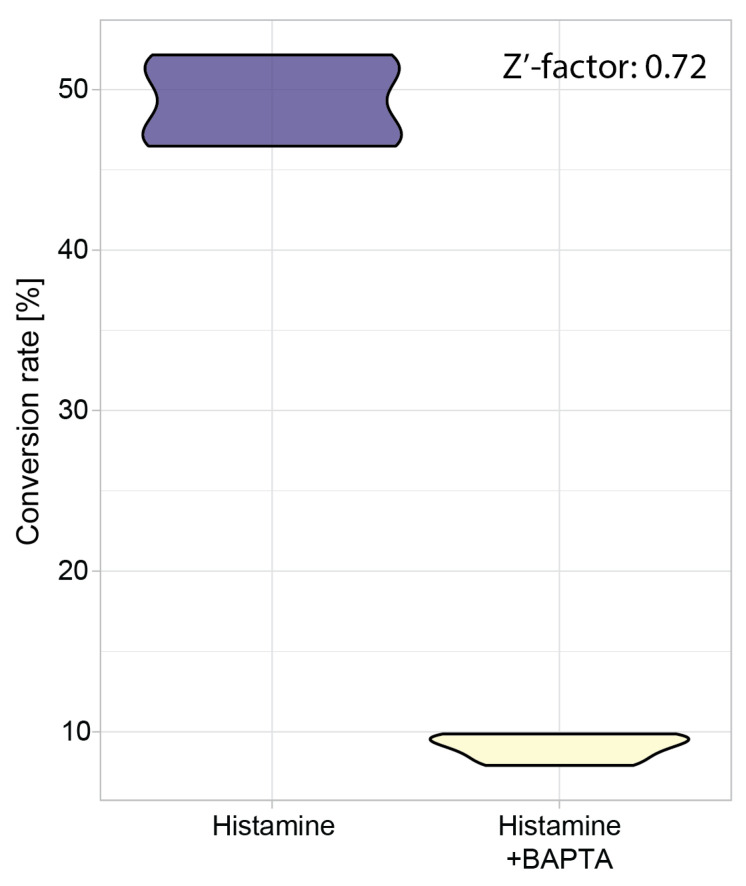
Z’-Factor analysis based on all four parabolas of the first parabolic flight day. CaMPARI2-F391W transfected human C28/I2 chondrocytes were treated with 100 µM histamine to induce photoconversion of CaMPARI2-F391W with 405 nm light. In addition, the Histamine + BAPTA condition was pretreated with 10 µM of the calcium (Ca^2+^) chelator BAPTA-AM as a negative control. The calculated Z’-factor for the first flight day is 0.72.

**Figure 9 biomedicines-10-00138-f009:**
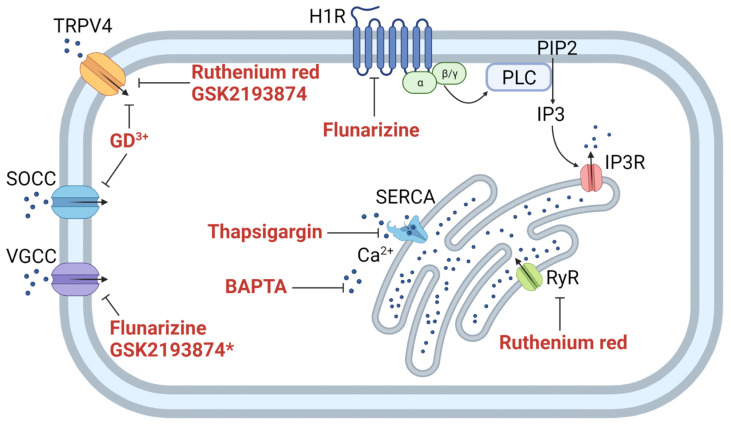
Transporters and receptors involved in calcium (Ca^2+^)-signalling and inhibitors used during the 75th ESA parabolic flight campaign. Blunted-head arrow show inhibiton of transporters and pumps by its associated drug, normal arrows show molecule movement through transporters or in signal cascades. H1R: Histamine H_1_ receptor; IP3: Inositol trisphosphate; IP3R: Inositol trisphosphate receptor; PIP2: Phosphatidylinositol 4,5-bisphosphate; PLC: Phospholipase C; RyR: Ryanodine receptors; SERCA: Sarco/endoplasmic reticulum Ca^2+^-ATPase; SOCC: Store-operated Ca^2+^ channel; TRPV4: Transient receptor potential cation channel subfamily V member 4; VGCC: Voltage-gated Ca^2+^ channel. This figure was created with BioRender.com (accessed on 5 January 2022).

**Table 1 biomedicines-10-00138-t001:** Drugs applied during the experiment.

Drug	Concentration	Solvent	Action	Reference
Untreated	-	-	-	
DMSO	0.1%	-	DMSO control	
Gadolinium (III) chloride (Gd^3+^)	10 µM	H_2_O	Blocker of extracellular calcium (Ca^2+^) entry	[38,40,41,46,49,59,60,61]
Thapsigargin	1 µM	0.1% DMSO	Blocker of SERCA (sarco/endoplasmic reticulum Ca^2+^-ATPase)	[47]
Flunarizine	50 µM	0.1% DMSO	Blocker of Ca^2+^ entry and histamine H1	[62,63]
Ruthenium red	10 µM	H_2_O	Blocker of CatSper1, KCNK3, RyR1, RyR2, RyR3, TRPM6, TRPM8, TRPV1, TRPV2, TRPV3, TRPV4, TRPV5, TRPV6, TRPA1, CALHM1, TRPP3, PIEZO	[64]
GSK2193874	10 µM	0.1% DMSO	Selective blocker of TRPV4	[65]
BAPTA-AM	10 µM	0.1% DMSO	Cell-permeable Ca^2+^-chelator	[66]

**Table 2 biomedicines-10-00138-t002:** Photoconversion metrics.

Metric	Definition
Conversion rate	The proportion of the green fluorescent cells that also display a red fluorescence (photoconversion)
Red objects mean fluorescence intensity	The mean red intensity of all cells
Merged objects red mean intensity	The mean red intensity of all cells displaying green and red fluorescence
Red-integrated intensity per well	The red intensity per well divided by the number of cells per well that display green and red fluorescence

## Data Availability

The quantitative imaging data and software codes (release 1.0) presented in this study are openly available in https://github.com/ZeroG-lab/PFCaMPARI at doi:10.5281/zenodo.5824491. Raw images are available on request from the corresponding author (M.B.).

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
