# Peer review of "Retrograde Analysis of Calcium Signaling by CaMPARI2 Shows Cytosolic Calcium in Chondrocytes Is Unaffected by Parabolic Flights"

_biomedicines, 2022, doi:10.3390/biomedicines10010138_

Round 1
Reviewer 1 Report
The submitted manuscript is a very interesting example of the article taking into consideration calcium signaling in chondrocytes exposed to parabolic flight maneuvers. Introduction part including description of calcium signaling and calcium indicators, especially CaMPARI technique, is very interesting and well prepared. Material and methods are well and properly described. How did the Authors manage to prevent from appearing of any air bubbles in the microplate wells under the lids before and during the flight? Was it monitored/checked during the flight? What kind of multiwall plates and adhesive films were used to prevent the medium leakage during the flight? – please add the necessary manufacturer information. It would recommend to describe it properly in the paper to make it possible to easily repeat such experiment by other experimentators.
The Results are properly described.
The Discussion section is prepared in a thoughtful way and the sufficient number of articles was cited. The study showed that calcium signaling by CaMPARI2 technique may be successfully applied for calcium imaging, even during parabolic flights, and cytosolic calcium in chondrocytes is unaffected by microgravity exposure. The Figure 9 is really helpful in data interpretation.
Summarizing, I recommend this manuscript for publication when the Authors correct the abovementioned issues in the article.
Author Response
We want to thank the reviewer for their time and their thoughtful comments. Here are our responses to the points raised during the review.
Point1: How did the Authors manage to prevent from appearing of any air bubbles in the microplate wells under the lids before and during the flight?
Response 1: Great care has been taken to fill wells of the microplates with cell culture medium completely. Next, the plates were sealed with adhesive film. After removing the microplates from the functional units, no air bubbles were detectable in the microplates. Furthermore, air bubbles would not have affected the illumination or microscopy of the adherent cells, which occurred from the bottom of the well. However, air bubbles can cause a calcium increase in contact with the cells, which we did not observe during the parabolic flight. Therefore, the manuscript has been reworded in the respective sections to reflect our efforts to prevent air bubbles.
Point 2: Was it monitored/checked during the flight?
Response 2: Air bubbles were not checked for during the flight as the microplates were in tight containers due to flight-safety reasons. However, no air bubbles were observed after applying adhesive film and after removal after the flight.
Point3: What kind of multiwall plates and adhesive films were used to prevent medium leakage during the flight?
Response3: Microplates and adhesive film were purchased from Greiner Bio-One (µCLEAR® BLACK, #655090; EASYseal™, #676001). This information is available in the manuscript now in sections 2.3 and 2.7.
Reviewer 2 Report
The authors introduce a new experimental setup based on the fluorescent Ca2+ reporter CaMPARI2 in parabolic flight conditions and then used histamine pathway inhibitors. They performed nice experiments, but they need to do the same experimental setup without the gravity conditions to be sure that the flight is responsible of these changes and also if the inhibition pathway effects is the same without gravity.
Minor concerns
- Order of figures should be made according to the order of appearance, for instance fig 2 should be 1
- Tukey posthoc test should reflects the differences between what groups
Author Response
We want to thank the reviewer for their time and their thoughtful comments. Here are our responses to the points raised during the review.
Point 1: They performed nice experiments, but they need to do the same experimental setup without the gravity conditions to be sure that the flight is responsible of these changes and also if the inhibition pathway effects is the same without gravity.
Response 1: The changes in gravity during the parabolic flight had no observable effect on intracellular calcium levels in chondrocytes under our experimental conditions. The purpose of the Histamine treatment post-flight was to test the cells' ability to photoconvert to a stimulus and the overall health of the chondrocytes after the whole length of the parabolic flight. Seeing that calcium levels in those chondrocytes could still be increased after the flight proved that the absence of a calcium increase in response to changes in gravity during the flight was not due to an inability of the cells to display an increase in calcium levels, but an actual absence of gravity-elicited calcium elevations.
The purpose of the pharmacological inhibitors was to further dissect the underlying signaling pathways in case of a gravity-induced increase in calcium levels. This, of course, proved futile after no calcium increase became apparent during the flight. The inhibitors did, however, influence the post-flight Histamin-signaling.
At this point, we have no reason to believe that the changes in gravity during the parabolic flight would cause a permanent alteration in the cells concerning the inhibitory effects of the pharmacological agents to Histamine. A prior experiment in our lab published by Andreas Hammer in his Master thesis [1] shows that Thapsigargin, Flunarizine, and Ruthenium red display the same inhibitory effect as seen after the parabolic flight (please see the attachment). However, the absolute conversion rates were lower since we optimized the transfection and treatment protocols for the parabolic flight and used prototype hardware for the preliminary experiments.
- Hammer, A. Establishing the genetically encoded calcium-sensing protein CaMPARI2 in human chondrocytes, Johann-Wolfgang-Goethe Universität, 2021.
Point 2: Order of figures should be made according to the order of appearance.
Response 2: We reordered all figures in the manuscript and moved two figures into the Material and Methods section to be more concise with the order of appearance in the text.
Point3: Tukey posthoc test should reflect the differences between what groups?
Response 3: An explanatory sentence was added to the figure legend of figures 5, 6, and 7.

Round 2
Reviewer 2 Report
Now it can be accepted